# Feasibility of Transcervical Robotic-Assisted Esophagectomy (TC-RAMIE) in a Cadaver Study—A Future Outlook for an Extrapleural Approach

**Peter Philipp Grimminger** [1,*], **Pieter Christiaan van der Sluis** [1], **Hubert Stein** [2], **Hauke Lang** [1], **Richard van Hillegersberg** [3] **and Jan-Hendrik Egberts** [4]

[1]  Department of General-, Visceral- and Transplant Surgery, Medical Center of the Johannes Gutenberg University, 55131 Mainz, Germany
[2]  Department of Global Clinical Development, Intuitive Surgical, Sunnyvale, CA 94086, USA
[3]  Department of Surgery, University Medical Center, 3584 CX Utrecht, The Netherlands
[4]  Department for General, Visceral, Thoracic, Transplantation and Pediatric Surgery, University Hospital Schleswig Holstein, 24105 Kiel, Germany
*   Correspondence: Peter.grimminger@unimedizin-mainz.de; Tel.: +49-613-117-2063/or -7291; Fax: +49-613-117-477-173

**Abstract:** In recent years, the evolution of advanced robotic medical systems has increased rapidly. These technical developments have led to advanced robotic systems, such as the da Vinci Xi, which allows superior controlled complex procedures and innovative surgical strategies. In esophageal surgery, the robotic-assisted minimally invasive esophagectomy (RAMIE) procedure is being developed and carried out with increasing frequency at centers worldwide. Recently, a new single port robotic system was introduced (da Vinci Single Port (SP)), which may allow for the exploration of new routes, such as transcervical robotic assisted minimally invasive esophagectomy (TC-RAMIE). This approach avoids opening the pleura by entering the mediastinum through the jugular window. In this report, we describe the technical steps of the TC-RAMIE using the new da Vinci SP system and compare it to the da Vinci Xi system.

**Keywords:** DaVinci; transcervical; robotic esophagectomy; esophageal cancer

## 1. Introduction

Most common robotic assisted procedures for esophageal surgery use either transhiatal or transthoracic approaches. The cervical access is commonly performed as open surgery when a cervical anastomosis or a three field lymphadenectomy is indicated. The cervical mediastinoscopic approach is not commonly used in minimally invasive esophageal surgery. However, the development of robotic systems, such as the da Vinci Xi and the new da Vinci Single Port (SP) (Intuitive Surgical Inc., Sunnyvale, CA, USA), has allowed for controlled transcervical extrapleural mediastinoscopic access, which may be a feasible approach for a certain group of patients. Potential clinical benefits include the avoidance of single lung ventilation, as well as the postoperative development of pneumonia, which is one of the most common morbidities after transthoracic esophageal surgery. Here, we describe the possible transcervical robotic approach using the new da Vinci SP system in comparison to the da Vinci Xi system.

## 2. Materials and Methods

This study is a pre-clinical cadaver study.

### 2.1. System and Instruments

Utilized were the DaVinci Xi System and the DaVinci SP System (both Intuitive Surgical, Sunnyvale, CA, USA) with their respective instrumentation.

### 2.2. Procedure Set Up

Three human cadaver models were used for this feasibility study in an institutional review board (IRB) approved clinical lab. The abdominal part has been described in previous publications and can be performed with similar techniques [1,2], using either a multiport (da Vinci XI) or a single port approach with two assistant trocars (da Vinci SP).

For the transcervical approach, the patient is in a supine position with the head reclined and turned to the right side. The da Vinci system (Xi/SP) is positioned at the right side of the patient. Transcervical esophagectomy was performed with the da Vinci Xi in one cadaver and with the da Vinci SP in two of the models.

### 2.3. Technique Description Transcervical Approach

A cervical incision of 3 cm is created on the left side of the neck for mediastinal preparation (Figure 1), as previously described for the da Vinci Xi system [3–6]. The mastoid muscle is lateralized, and an x-small Alexis Wound Retractor (Applied Medical, Rancho Santa Margarita, CA, USA) is inserted. Next, the esophagus is prepared and looped using a silicone sling, which is used for retraction of the esophagus during the upper mediastinal preparation. A Gel Point mini (Applied Medical, Rancho Santa Margarita, CA, USA) is placed inside the Alexis Wound retractor but not tightened down to the skin, to create a sterile tubular access to the neck.

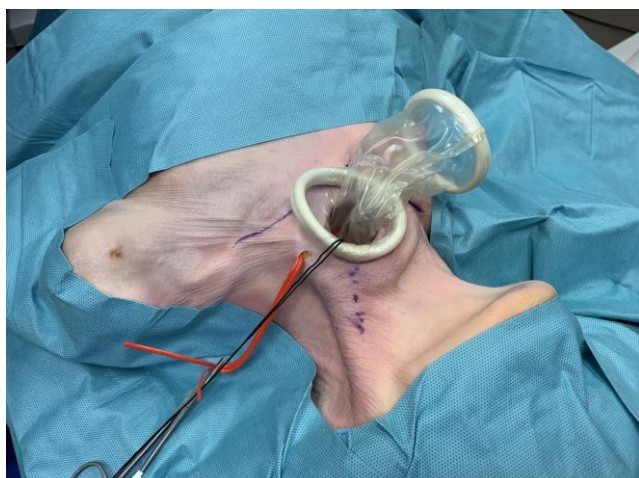

**Figure 1.** Transcervical incision at neck with the inserted x-small Alexis Wound Retractor.

For the da Vinci SP system, the one inch SP trocar is inserted into the Gel-Port and connected to the sleeve, as shown in Figures 2a and 3a. This set up is comparable to the previously described set up [7] for the da Vinci Xi system, where three 8 mm trocars are inserted into the Gel-Port in a triangulated position and connected to the sleeve (Figures 2b and 3b). Using low carbon dioxide insufflation up to 5 mmHg, a capnomediastinum is created, and the robot trocars are connected to their respective da Vinci system (Figure 3a,b). Using an air seal (SurgiQuest, Inc., Utica, NY, USA) may be useful to avoid "fluttering" of the tissue in the small surgical workspace, as well as a bilateral chest drain to avoid a pneumothorax.

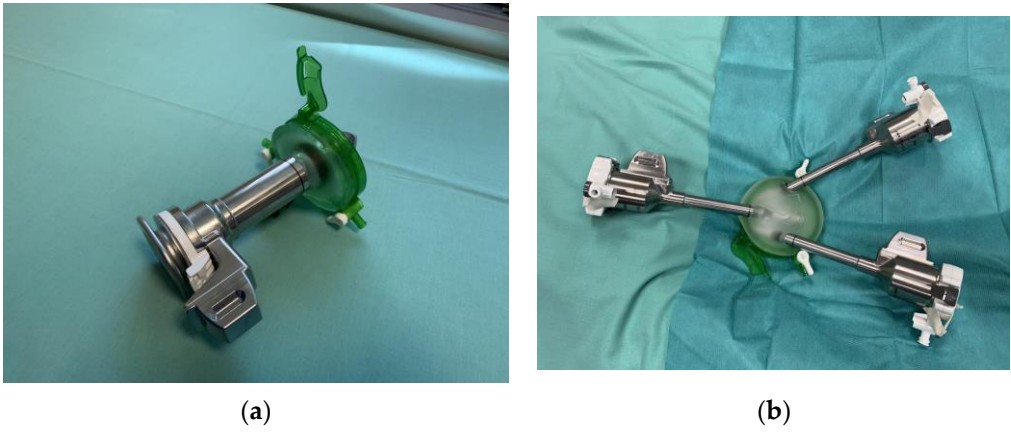

(**a**)　　　　　(**b**)

**Figure 2.** Trocar set up for da Vinci (Single Port) SP (**a**) and da Vinci Xi system (**b**) using a Gel Port.

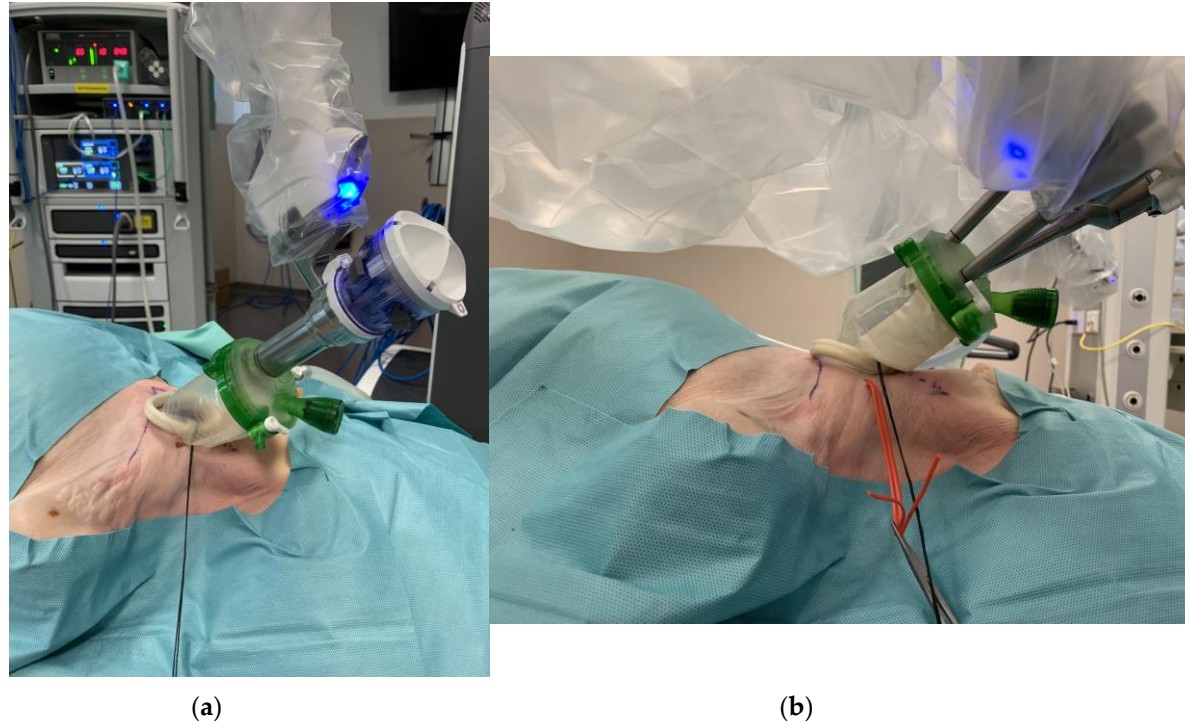

(**a**)　　　　　(**b**)

**Figure 3.** da Vinci SP (**a**) and Xi system (**b**) docked to the sleeve on the left side of the neck.

For the da Vinci SP, a newly designed steerable endoscope (steerable tip in multiple directions), monopolar scissors, fenestrated bipolar, and round tooth forceps (all da Vinci SP) were used for the preparation of the upper mediastinum. The left recurrent nerve is one of the first structures to be identified between the esophagus and the left tracheal wall (Figure 4). Then, the dissection of the upper esophagus and its surrounding tissue, including the lymph nodes along the left recurrent nerve, are dissected. The esophagus can be detached from the pars membranacea using monopolar scissors. During esophageal preparation, the spine is used lateral and ventral landmark for continuous dissection. Following these planes, the parietal pleura can be identified and should be followed as the appropriate border for further dissection.

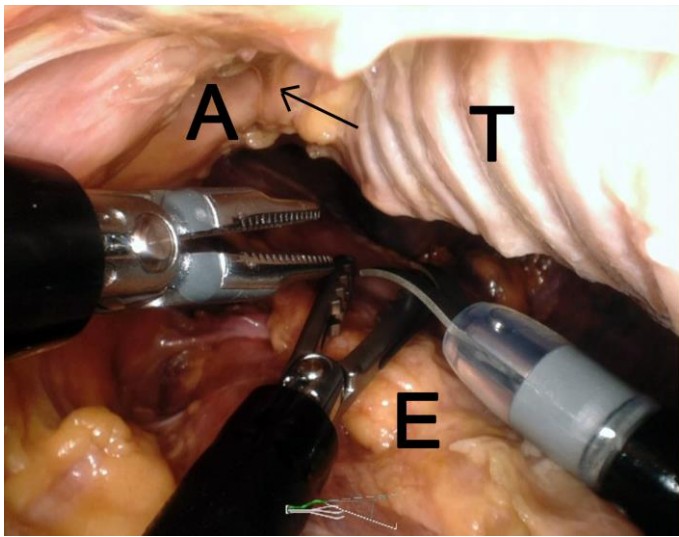

**Figure 4.** Paratracheal view of the left side of the trachea (T) in the upper mediastinum. (E) esophagus; (A) aortic arch; arrow points to right recurrent nerve.

Next, the aortic arch is encountered, and the left recurrent nerve is identified and followed for lymph node and tissue dissection. The esophagus and para-esophageal tissue are then circumferentially freed. The azygos vein and the aorta are used as landmarks for preparation (Figure 5). Retracing the esophagus upwards and backwards at the same time, the trachea can be followed to the carina, where the right and left main bronchus are identified and dissected. Then, using the da Vinci SP system, the carinal lymph node dissection is greatly facilitated from this transcervical access. Further mobilization of the esophagus inferiorly was continued with the SP system until the crus of the diaphragm was reached (Figure 6). Any aortoesophageal perforators encountered were ligated using clips combined with monopolar or bipolar instrumentation (Figure 7). With the Xi system inferior dissection was continued to a point until limitations in reach were hindering effective surgical performance, as judged by the operators.

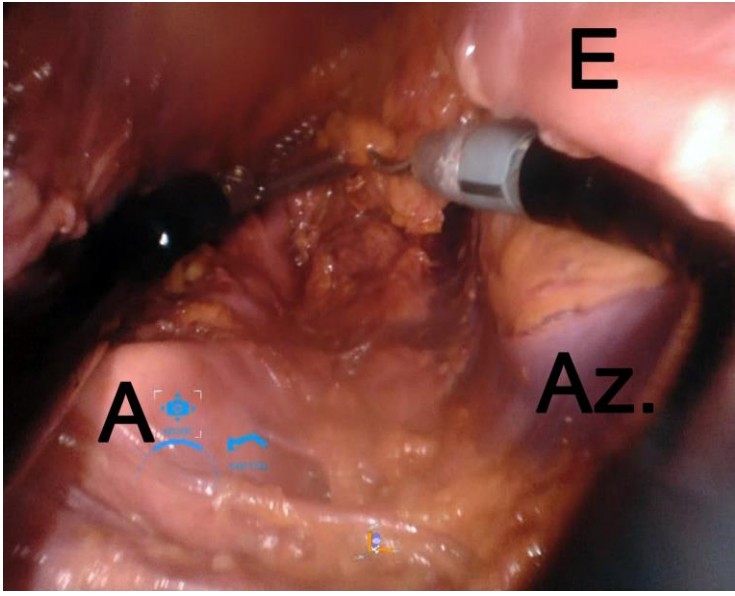

**Figure 5.** Da Vinci SP approach into the mid and lower mediastinum; esophagus (E); azygos vein (Az.); aorta (A).

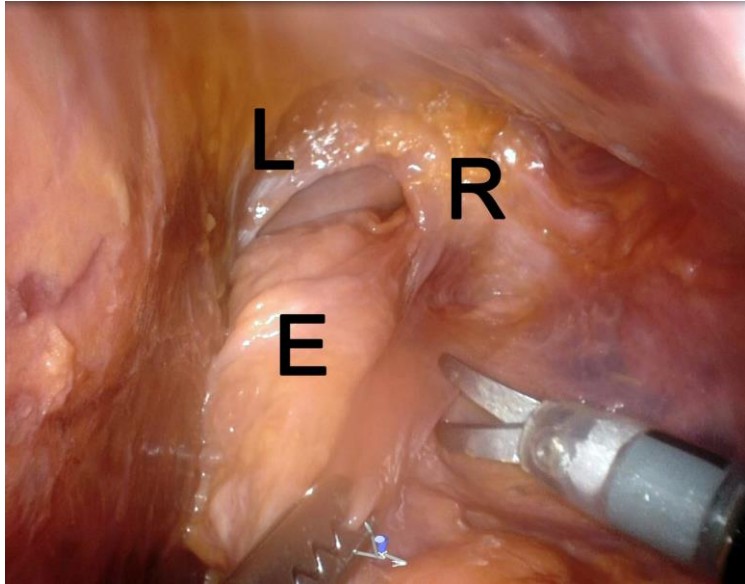

**Figure 6.** Da Vinci SP transcervical view on the hiatus; (L) left crus; (R) right crus; (E) esophagus.

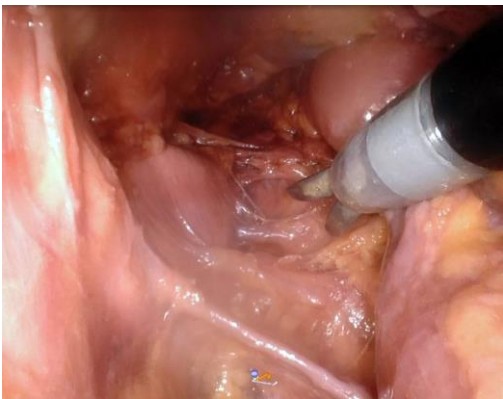

**Figure 7.** Da Vinci SP view on the aortal branches in the lower mediastinum.

## 3. Reconstruction

The specimen was resected after de-docking the robot. The gastric conduit was created transabdominally and robotically with a 45 mm Endo Wrist Stapler (green and blue reloads) on the DaVinci Xi and with a laparoscopic stapler on daVinci SP. A drain was attached to the gastric conduit and pulled up with a laparoscopic grasping instrument. The cervical anastomosis could then be performed using commonly described techniques with hand-sewn, side-to-side stapled, or circular stapled, anastomosis. We prefer the Orringer side-to-side stapled anastomosis [7]. Afterwards, the incision was closed and drainage could be optionally placed.

## 4. Results

The mobilization of the esophagus was successfully completed in all three cadaver models (two with the da Vinci SP and one with the da Vinci Xi). The da Vinci SP system allowed dissection of the esophagus until the hiatal crus was reached without interference of the instrumentation in two cadavers. In the transcervical esophagectomy performed with the da Vinci Xi, we were unable to reach the hiatus. Safe manipulation of the Xi System via the transcervical route was possible until the carina. Below the level of the internal carina, external interferences between the robotic arms and patient anatomy were encountered at a higher frequency, letting us determine the continuation of the dissection to no longer be safe or effective.

## 5. Discussion

The described robotic-assisted transcervical MIE technique is an approach whose feasibility, reproducibility, and clinical value for certain patient groups must be evaluated in further studies. Possible advantages could include the controlled dissection of high esophageal tumors, as well as superior left recurrent lymph node dissection and reduction of pulmonary complications due the complete extra-pulmonary dissection. The da Vinci SP platform seems to have advantages over the da Vinci Xi system, especially in the dissection of the mid to lower mediastinum via the transcervical approach. Advantages of the da Vinci SP are less interferences and better control of the four combined instruments delivered through a small lumen trocar. Furthermore, the smaller diameter instruments of the SP (6 mm) allow for enhanced maneuverability in the mediastinal space.

**Author Contributions:** Conceptualization, methodology, writing—original draft preparation, writing—review and editing and supervision: P.P.G., P.C.v.d.S., H.S., H.L., R.v.H. and J.-H.E.

**Conflicts of Interest:** Hubert Stein is employed by Intuitive Surgical. Peter Philipp Grimminger, Richard van Hillegersberg and Jan-Hendrik Egberts are proctors for Intuitive Surgical. All other authors do not have conflict of interest.

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
