# Peer review of "Feasibility of Transcervical Robotic-Assisted Esophagectomy (TC-RAMIE) in a Cadaver Study—A Future Outlook for an Extrapleural Approach"

_applsci, doi:10.3390/app9173572_

Round 1

Reviewer 1 Report

the authors have to be congratulated for an undertaking that will help esophagectomies become less invasive in an intent to have less morbidity. Yet, in live human beings, it will require strong nerves, especially in view of peritumoral inflammation or preoperative understaging. 

one remark/question: is it correct that you didn't reach the hiatus with the Xi-System? how far did you get? please elaborate more on that

Author Response

Reviewer 1:

One remark/question: is it correct that you didn't reach the hiatus with the Xi-System? How far did you get? please elaborate more on that

Response Reviewer 1:

Safe manipulation of the Xi System via the transcervical route was possible until the carina. Below the level of the carina internal and external interferences between the robotic arms and patient anatomy were encountered in higher frequency letting us deem the continuation of the dissection not safe and effective anymore. This was added in track changes to the result section.

Reviewer 2 Report

I appreciate the opportunity to review this manuscript. It is well written and of interest to a fair number of readers.

Can the authors give more information on how the conduit was created and pulled up to the neck for anastomosis?

Can the authors comment on what energy was used to ligate the aortoesophageal perforators?

Author Response

Reviewer 2:

Can the authors give more information on how the conduit was created and pulled up to the neck for anastomosis?

Can the authors comment on what energy was used to ligate the aortoesophageal perforators?

Response Reviewer 2

The gastric conduit was created transabdominally robotically with a 45 mm EndoWrist Stapler (green and blue reloads). A drain was attached to the gastric conduit and pulled up with a laparoscopic grasping instrument.

The aortoesophageal perforators were ligated using clips combined with the monopolar or bipolar instrumentation on the SP system.

This was added with track changes to the manuscript.